# G-quadruplexes formed by Varicella-Zoster virus reiteration sequences suppress expression of glycoprotein C and regulate viral cell-to-cell spread

**Woo-Chang Chung[1]◉, Subramaniyam Ravichandran[2]◉¤, Daegyu Park[1], Gwang Myeong Lee[1], Young-Eui Kim[1], Youngju Choi[1], Moon Jung Song[3], Kyeong Kyu Kim[2,4], Jin-Hyun Ahn◉[1,4]\***

1 Department of Microbiology, Sungkyunkwan University School of Medicine, Suwon, Republic of Korea,
2 Department of Precision Medicine, Sungkyunkwan University School of Medicine, Suwon, Republic of Korea, 3 Department of Biotechnology, College of Life Sciences and Biotechnology, Korea University, Seoul, Republic of Korea, 4 Biomedical Research Institute, Samsung Medical Center, Seoul, Republic of Korea

◉ These authors contributed equally to this work.
¤ Current address: Department of Biology, Stanford University, Stanford, California, United States of America
\* jahn@skku.edu

**Data Availability Statement:** All relevant data are within the manuscript and its Supporting information files.

## Abstract

G-quadruplex (G4) formed by repetitive guanosine-rich sequences plays important roles in diverse cellular processes; however, its roles in viral infection are not fully understood. In this study, we investigated the genome-wide distribution of G4-forming sequences (G4 motifs) in Varicella-Zoster virus (VZV) and found that G4 motifs are enriched in the internal repeat short and the terminal repeat short regions flanking the unique short region and also in some reiteration (R) sequence regions. A high density of G4 motifs in the R2 region was found on the template strand of ORF14, which encodes glycoprotein C (gC), a virulent factor for viral growth in skin. Analyses such as circular dichroism spectroscopy, thermal difference spectra, and native polyacrylamide gel electrophoresis with oligodeoxynucleotides demonstrated that several G4 motifs in ORF14 form stable G4 structures. In transfection assays, gC expression from the G4-disrupted ORF14 gene was increased at the transcriptional level and became more resistant to suppression by G4-ligand treatment. The recombinant virus containing the G4-disrupted ORF14 gene expressed a higher level of gC mRNA, while it showed a slightly reduced growth. This G4-disrupted ORF14 virus produced smaller plaques than the wild-type virus. Our results demonstrate that G4 formation via reiteration sequences suppresses gC expression during VZV infection and regulates viral cell-to-cell spread.

## Author summary

G-quadruplexes (G4s) are found in herpesvirus genomes. However, whether G4 formation regulates the growth and pathogenesis of Varicella-Zoster virus (VZV), which causes

**Funding:** This work was supported by grants from the National Research Foundation of Korea (NRF) funded by the Ministry of Science and ICT (2020R1A4A1018019, 2021M3A9I2080488, and 2022R1A2C1006748) to JHA. The funders had no role in study design, data collection and analysis, decision to publish, or preparation of the manuscript.

**Competing interests:** The authors have declared that no competing interests exist.

chickenpox in children and shingles in older adults, is unclear. Here, we identified ~150 putative G4-forming sequences (G4 motifs) in the VZV genome and found that these sequences are enriched in the repeat regions in the genome. In particular, the G4 motifs found in the R2 reiteration region are located on the template strand of ORF14 that encodes glycoprotein C (gC). We demonstrate that several G4 motifs in R2 form anti-parallel G4 structures and that the formation of these G4s suppresses gC expression at the transcription level during virus infection. We also show that the G4-mediated attenuation of gC expression promotes viral spread to neighboring cells. Therefore, VZV has evolved to control gC expression through G4 formation by reiteration sequences for regulation of viral cell-to-cell spread.

## Introduction

Repetitive guanosine-rich sequences connected by short stretches of nucleotides in single-stranded DNA or RNA can fold into a distinct type of tertiary structure known as a G-quadruplex (G4). In a G4 structure, four guanine bases are connected by Hoogsteen bonds to form a G-tetrad; multiple G-tetrads stack on top of each other, resulting in various types of G4 with different numbers of G-tetrads. G4s are stabilized by several monovalent and divalent cations including $K^+$ and $Na^+$. G4s can be produced within a single nucleic acid molecule to form intramolecular G4s or with several molecules to produce intermolecular G4s. G-runs in a G4 structure can be parallel or antiparallel [1,2]. G4s have been shown to play crucial roles in diverse biological processes, such as telomere protection from nuclease attack, regulation of gene transcription and translation, and regulation of DNA or RNA replication and genome stability [3,4].

The *Herpesviridae* family consists of enveloped viruses with a long linear double-stranded DNA genome. There are eight human herpesviruses and these are sub-classified as alpha-, beta-, and gamma-herpesviruses. Genome-wide bioinformatics analyses revealed that the herpesvirus genomes have high densities of G4-forming sequences (also called G4 motifs) [5,6]. Several studies have indicated the functional roles of G4s in the herpesvirus life cycle. G4 formation in the promoter regions of herpes simplex virus-1 (HSV-1), human cytomegalovirus (HCMV), and Kaposi's sarcoma-associated herpesvirus (KSHV) has been shown to regulate viral gene expression [7–10]. G4 formation in Epstein-Barr virus (EBV) and KSHV mRNAs downregulates mRNA translation, suppressing antigen presentation [11–14]. In addition, several G4-binding ligands can inhibit herpesvirus growth [9,15,16]. Therefore, G4 formation in the herpesviral genomes is thought to affect diverse steps that are critical in viral lytic and latent infection.

Varicella-Zoster virus (VZV) belongs to the *alpha-herpesvirinae* subfamily and has an average 125-kb genome size with at least 71 open reading frames (ORFs). VZV can cause chickenpox (varicella), an early childhood disease [17]. After primary infection, VZV establishes lifelong latent infection in sensory trigeminal and dorsal root ganglia. In aged people, latent VZV can reactivate, often causing shingles (herpes zoster) in the skin [18,19]. The most common complication of herpes zoster is chronic nerve pain (called postherpetic neuralgia) [20] and about 25% of immunocompetent adults older than 50 years experiences herpes zoster-related complication [21]. Although VZV vaccines can reduce varicella and herpes zoster effectively [22], the number of VZV global incident case has increased, with more than 83 million new VZV infection cases in 2019 [23]. As elderly population worldwide has been increased, herpes zoster is a new global health burden.

Like HSV-1, the VZV genome consists of unique and repeat regions. The ~100-kb unique long ($U_L$) region is flanked by the terminal repeat long ($TR_L$) region and the internal repeat long ($IR_L$) region and is followed by the 5.2-kb unique short ($U_S$) region flanked by the internal repeat short ($IR_S$) region and the terminal repeat short ($TR_S$) region [17,24]. The VZV genome shows a unique feature having five GC-rich reiteration (R) sequence regions that have tandemly repeated units (called elements) that vary among strains [25,26]. A previous study on the distribution of G4 motifs in the VZV genome demonstrated enrichment of putative G4 motifs in the regulatory regions and the repeat regions [5]. Another study reported that the VZV immediate early (IE) promoters contain multiple G4 sequences that may have regulatory roles for VZV lytic replication [7]. However, studies on how G4 formation in the VZV genome regulates the viral life cycle are limited, especially in the context of virus infection.

In this study, we searched for G4 motifs in the entire VZV genome using *in silico* analysis and identified ~150 putative G4 motifs. These sequences were enriched in the $IR_S$ and $TR_S$ regions and were also found in some of the R regions. We found a high density of G4 motifs in R2. These sequences are located on the template strand of ORF14 encoding glycoprotein C (gC), which acts as a virulence factor [27,28]. We performed biophysical analyses to characterize G4 formation from the predicted sequences in R2. We investigated whether G4 formation in R2 regulates gC expression by generating the mutant ORF14 genes in which the G4-forming potential was disrupted. The role of G4s was also investigated by generating the recombinant viruses containing the G4-disrupted mutant ORF14 gene and its revertant gene. Finally, we investigated whether alteration of gC expression by G4s can regulate viral cell-to-cell spread.

## Results

### Genome-wide analysis of G4 motifs in the VZV genome

The 124 kb genome of the VZV Dumas strain (NC_001348.1), a reference strain of VZV, was mined for putative G4 motifs using the Quadparser program. The prediction identified 148 putative G4 motifs (denoted as GQ1 to GQ148) in the Dumas genome (S1 Appendix). We divided the predicted sequences into three main categories based on the loop architecture: conventional, long-looped, and bulged G4s. Conventional G4s were characterized by loops of 1–7-nt length and four complete G-tracts (3 G-runs). We predicted 14 conventional GQs in the entire genome. In the case of long-looped G4s, we restricted the loop length of two loops to 1–7 nt, whereas the length of the third loop was allowed to be 8–30 nt. We predicted 84 long-looped GQs in Dumas.

Several studies have indicated that the stability of the G4 is inversely proportional to the loop length of the G4s [29,30]. The optimal loop length for stable G4 formation was predicted to be 10 nt long, but subsequent studies showed that G4s with longer loop lengths are also biologically relevant and should be considered for functional studies [31–34]. We also considered including bulged G4s in our study, since previous studies indicated that sequences could form stable G4s even in the presence of an incomplete G-tract [35]. The nucleotide between contiguous Gs in a G-tract can bulge outwards and accommodate the adjacent guanine to complete the G-tract with a bulge [35]. For consideration of G4 stability, only G4 motifs containing a single bulge were considered. In our computational study, we predicted 50 bulged GQs in the Dumas genome.

In addition to the conventional, long-looped and bulged G4s, G4 formation is also observed in sequences that do not contain loops between the G-tracts. The majority of G4s conform to the conventional, bulge, and long-loop G4 categories, but examples of G4s without loops have been observed in VZV ORF62/63 [7]. Additionally, in the case of the bulged G4s, the position of the bulge can alternate between the first or second G in a G-tract and the second and third

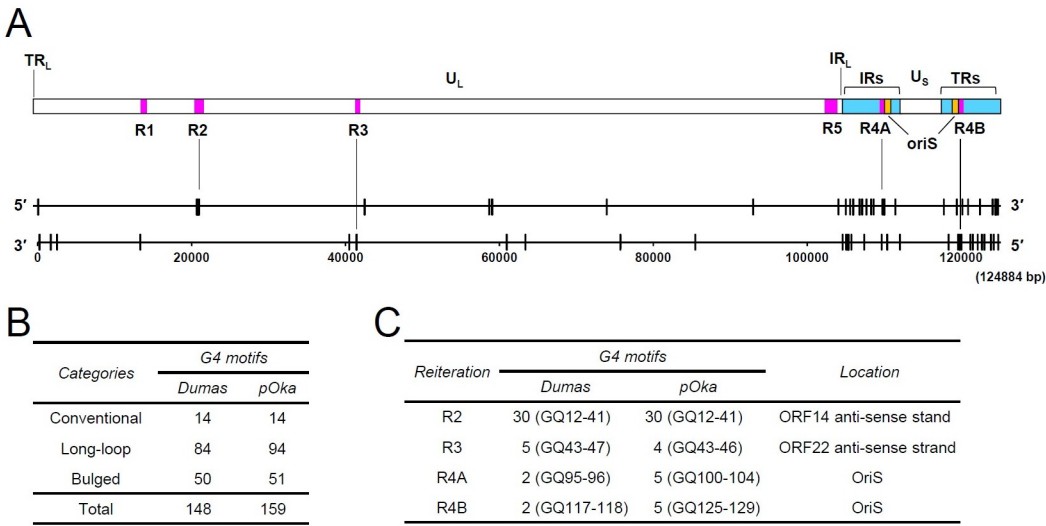

**Fig 1. Distribution of putative G4 motifs in the VZV genome.** (A) The locations for the predicted 148 G4 motifs (denoted as GQ1 to GQ148) in the Dumas strain are indicated as short vertical lines on the top and bottom sequences of the viral genome. The locations of the terminal repeat long (TR$_L$), unique long (U$_L$), internal repeat long (IR$_L$), internal repeat short (IR$_S$), unique short (U$_S$), terminal repeat short (TR$_S$), oriS and reiteration sequences (R1, R2, R3, R4A, R4B, and R5) are also indicated. (B and C) The numbers of predicted G4-motifs in different categories (B) and in different reiteration sequence regions (C) in Dumas and pOka strains are summarized.

G. In the case of Dumas, we observed that the sequence prediction was same in both bulge positions; therefore, we retained the sequences observed from one schema.

Our *in silico* analysis revealed that G4 motifs are highly enriched in two long repeated sequences, the IR$_S$ and the TR$_S$ regions (Fig 1A). The VZV genome contains five reiteration sequences (designated to R1, R2, R3, R4A, R4B, and R5). We found that several G4 motifs are located within the R2, R3, R4A, and R4B regions (Fig 1A). We also predicted the putative G4 motifs in the pOka strain of VZV (AB097933.1) to compare with the Dumas strain. pOka is a parental strain of vOka, a licensed vaccine strain [36], and the pOka-derived bacmid is wildly used for analysis of VZV gene functions [37]. We predicted 159 G4 motifs in the pOka genome. The number of conventional G4 in pOka was the same as that in Dumas, although the numbers for long-loop and bulged G4s were slightly different (94 and 51 in pOka, respectively) due to the sequence differences between these two strains (Fig 1B and S1 Appendix). The pOka genome also showed similar enrichment of G4 motifs in the IR$_S$ and the TR$_S$ regions and in R2, R3, R4A, and R4B (Fig 1C and S1 Appendix).

## Distribution of G4 motifs in IR$_S$

IR$_S$ contains ORF62, OriS, ORF63, and ORF64. TR$_S$, which contains ORF69, ORF70, OriS, and ORF71, is an inverted sequence of IR$_S$. The G4 motifs in IR$_S$ are found upstream of ORF61, in the sense- and non-sense strands of ORF62, and in OriS, which overlaps with the bi-directional promoter for ORF62 and ORF63. The positions of G4 motifs in IR$_S$ are indicated in S1 Fig. We found that GQ65 lies in the minimal ORF61 promoter activated by the ORF62 transactivator, with overlapping with two non-canonical Sp1-binding sites, which are required for effective ORF61 expression [38]. The ORF62/63 intergenic region containing OriS acts as a bi-directional promoter for ORF62 and ORF63 transcription [39]. Several G4 motifs are found in this intergenic region. GQ94 overlaps with the promoter region of ORF62 [40], while GQ103 and GQ104 are located in the promoter of ORF63 [41]. Besides the R4A reiteration

sequence, the OriS region contains three consensus binding sites (Boxes A, B, and C) for the VZV origin-binding ORF51 protein and a neighboring 46-bp AT-rich palindrome sequence [42–44]. Notably, GQ95 and GQ96 are found in R4A, and GQ97 and several G4 motifs overlapping with GQ97 are localized between ORF51-binding sites [45] (S2 Fig).

## Distribution of G4 motifs in R2

We focused on the G4 motifs in the R2 reiteration region since R2 shows a relatively high level of nucleotide variation among different strains [25,26] and is located within ORF14 encoding the glycoprotein C (gC). Most of the nucleotide variations observed in pOka and the other two clinical isolates YC01 and YC02, which were previously isolated from the patients of herpes zoster and chickenpox, respectively [46], were mapped in R2 (Fig 2A). The predicted G4

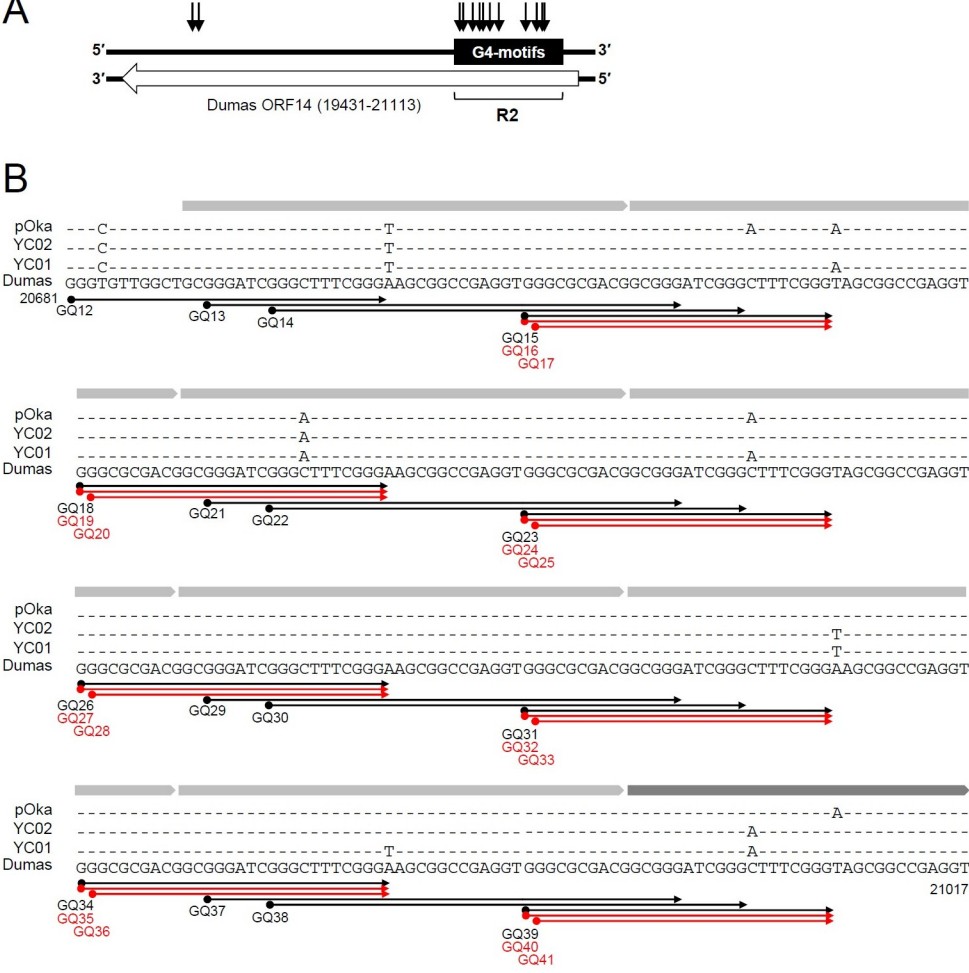

**Fig 2. Putative G4 motifs in R2 within ORF14.** (A) The position of the ORF14 coding sequence is indicated as an open arrow on the bottom strand of the VZV Dumas genome and the region containing putative G4 motifs in R2 is indicated as a black box on the top strand. The variations found in ORF14 of pOka, YC01, or YC02 compared with Dumas are indicated as black vertical arrows. (B) The G4 motifs (GQ12 to GQ41) found in R2 are indicated below the genome sequence of Dumas (from 20681 to 21017). The variations found in pOka, YC01, and YC02 are indicated above the Dumas sequences. All G4 motifs of R2 are predicted on the anti-sense strand of ORF14. Black arrows, GQs for long-looped G4; red arrows, GQs for bulged G4. Seven 42-bp elements and one 32-bp element in R2 are indicated as light and dark grey bars, respectively, above the sequences.

motifs in ORF14 (denoted as GQ12 to GQ41 in Dumas) were long-loop or bulge types and found on the template strand of ORF14 (Fig 2A). R2 consists of seven consecutive 42-bp elements and one final 32-bp element in these VZV strains. Based on the locations of G4 motifs, the R2 region could be divided into four segments, with each segment containing separated G4 motifs and G4 motifs overlapping them (Fig 2B). When sequences in pOka, YC01, and YC02 were aligned with Dumas, nucleotide variations found in R2 appeared to occur adjacent to G-runs in G4 motifs (Fig 2B). We performed a conservation analysis on 141 R2 sequences available in GenBank. Consistent with the result in Fig 2B, nucleotide variations in R2 largely occurred just downstream of G-runs in G4 motifs (S3 Fig and S2 Appendix).

## Biophysical analysis of G4 formation by R2 G4 motifs

To investigate the G4 formation in R2 and its role in ORF14 gene expression, we used sequences predicted from pOka since the pOka bacmid was available for mutational analysis. The same numbers of G4 motifs (also named as GQ12 to GQ41) were predicted at the same position in R2 of pOka compared with those in Dumas (Fig 2 and S1 Appendix). We synthesized the oligodeoxynucleotides (ODNs) corresponding to the identified sequences and their mutant sequences, in which three or five G-to-T or G-to-A changes are introduced in G-runs to disrupt G4 formation without affecting coding potential (S1 Table). The sequences of GQ15, GQ16, GQ18, GQ19, GQ23, and GQ24 are identical. However, we tested two different mutant sequences, GQ15m/16m/18m/19m and GQ23m/24m, not to disrupt the coding potential for ORF14. We also used these two mutant sequences to make the mutant ORF14 gene for later gene expression studies. Likewise, two different mutant sequences, GQ17m/20m and GQ25m, were analyzed for identical GQ17, GQ20, and GQ25 G4 motifs. Circular dichroism (CD) spectroscopy analysis of ODNs suggested that nine GQ motifs, except GQ21 and GQ22, might form anti-parallel G4s, which displayed a prominent peak at ~290 nm and a trough at ~260 nm in the presence of 100 mM KCl as reported previously [47–49]. The GQ21 and GQ22 ODNs appeared to fold into an unusual structure, with a ~280 nm peak and ~245 nm trough, which is different from that of a single-stranded 24mer poly(T) [9]. The CD spectra of all GQ mutants, except GQ22m, were significantly changed from those of wild-type sequences, suggesting sequence-specific formation of G4s by most G4 motifs predicted in R2, except GQ21 and GQ22 (Fig 3).

The formation of G4s was also investigated by measuring the thermal difference spectra (TDS) that are calculated as the difference between the UV spectra at the maximum unfolding temperature (90˚C) and the folding temperature (20˚C). G4s are known to show the characteristic TDS spectra with a negative peak at ~295 nm and positive peaks at ~240 and ~270 nm [50]. We found that among the nine GQ motifs that showed the CD spectra of G4s, eight G4 motifs, except GQ17, showed the TDS spectra of G4 (Fig 4A), while their mutant sequences only showed the major peak at ~270 nm, indicating the disruption of the G4 structure (Fig 4B). The CD and TDS results suggest that GQ12, 13, 14, 15, 26, 28, 29, and 30 motifs form a G4 structure.

To further analyze G4 structures formed by GQ12, 13, 14, 15, 26, 28, 29, and 30 motifs, we used a well-known G4-binding ligand Pyridostatin (PDS) (Fig 5A), which has previously been shown to have high affinity for the anti-parallel G4 structure [51–53]. We tested a minimum 1:2 stoichiometry of the ligand to consider the binding of the ligand on the G4 structure. Treatment with 2× molar ratio of PDS did not show any shifts in the ~290 nm peak for all G4s in the CD spectra, indicating that the topology of these G4s is not affected by PDS. We next investigated the stability of G4s by analyzing their CD thermal melting curves. The ellipticity at 290 nm, a wavelength attributed to the formation of the anti-parallel G4, was monitored in the

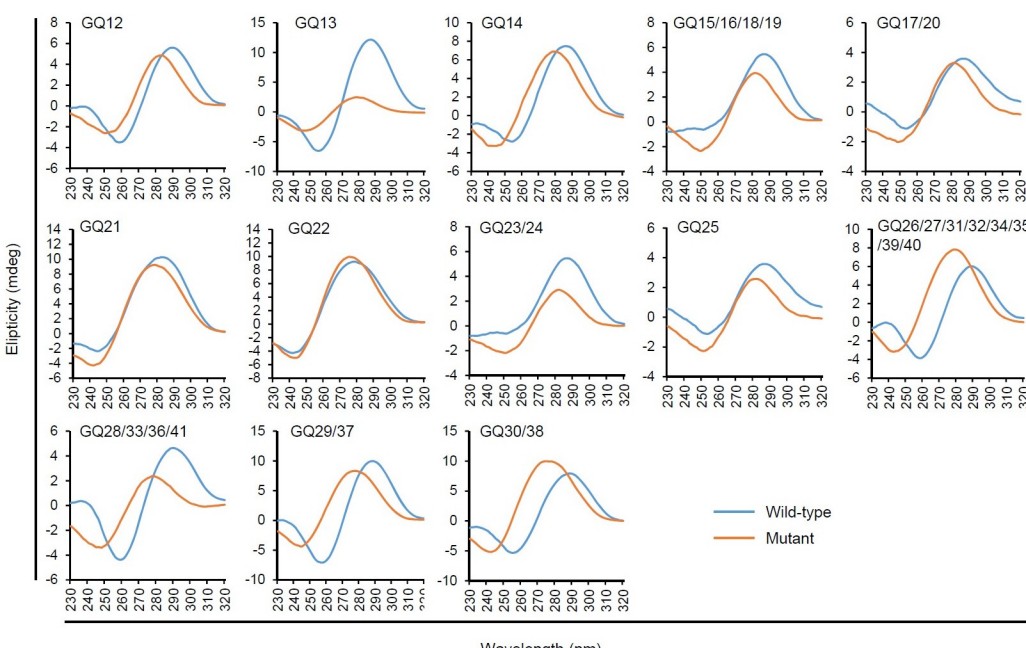

**Fig 3. CD spectroscopy analysis of ORF14 GQ ODNs from pOka.** CD spectra of wild-type and mutant ORF14 GQ ODNs (15 μM) from pOka annealed in 10 mM Tris-HCl pH 7.5 and 100 mM KCl are shown. Each spectrum was an average of three accumulations in the wavelength range between 230–320 nm. The spectra were blanked with buffer only. GQ15/16/18/19 and GQ23/24 are same sequences, but their mutant sequences are different. Similarly, GQ17/20 and GQ25 are the same sequences, but their mutant sequences are different. The data were normalized using the maximum ellipticity and smoothed using GraphPad Prism 5.

temperature range of 20–90˚C. All eight G4 motifs displayed conventional sigmoid curves with Tm values of 52.2–58.0˚C (Fig 5B and Table 1). The Tm values of the ORF14 G4s were increased by 0.5 to 7.5˚C in the presence of PDS (Fig 5B and Table 1). The thermal stability appeared to be determined by sequences rather than G4 structures. Among the eight G4 motifs that showed formation of a G4 structure in both CD and TDS analyses, the sequences of GQ12, 13, 15, 26, 28, and 29 appear to form a G4 structure whose Tm can be substantially increased (>3.0˚C) by PDS (Table 1).

ODNs with an intramolecular G4 structure are known to be more compact and migrate faster than unstructured mutants in a native polyacrylamide gel electrophoresis (PAGE) [54–56]. Therefore, we tested whether the G4 motifs identified to form a G4 structure can form a compact intramolecular structure in a native PAGE. When the wild-type and mutant GQ12, 13, 14, 15, 26, 28, 29, and 30 motifs were annealed in 100 mM KCl buffer and subjected to a native PAGE, most of wild-type ODNs, except GQ26 and GQ28 motifs, migrated faster than their mutant ODNs, while the difference of mobility between wild-type and mutants became diminished under denaturing conditions (with 7M Urea) (Fig 6). These results suggest that the GQ12, 13, 14, 15, 29, and 30 motifs can form a compact intramolecular structure.

## G4-mediated suppression of ORF14 gene expression in transfection assays

Having confirmed the G4 formation by several G4 motifs, which are distributed throughout the R2, we next asked whether G4s regulate ORF14 gene expression. To address this question, the G4-disrupted ORF14 mutant DNA (denoted as ORF14-G4m), in which all G4 motifs were replaced with the mutant sequences used for *in vitro* analysis (in S1 Table), was synthesized

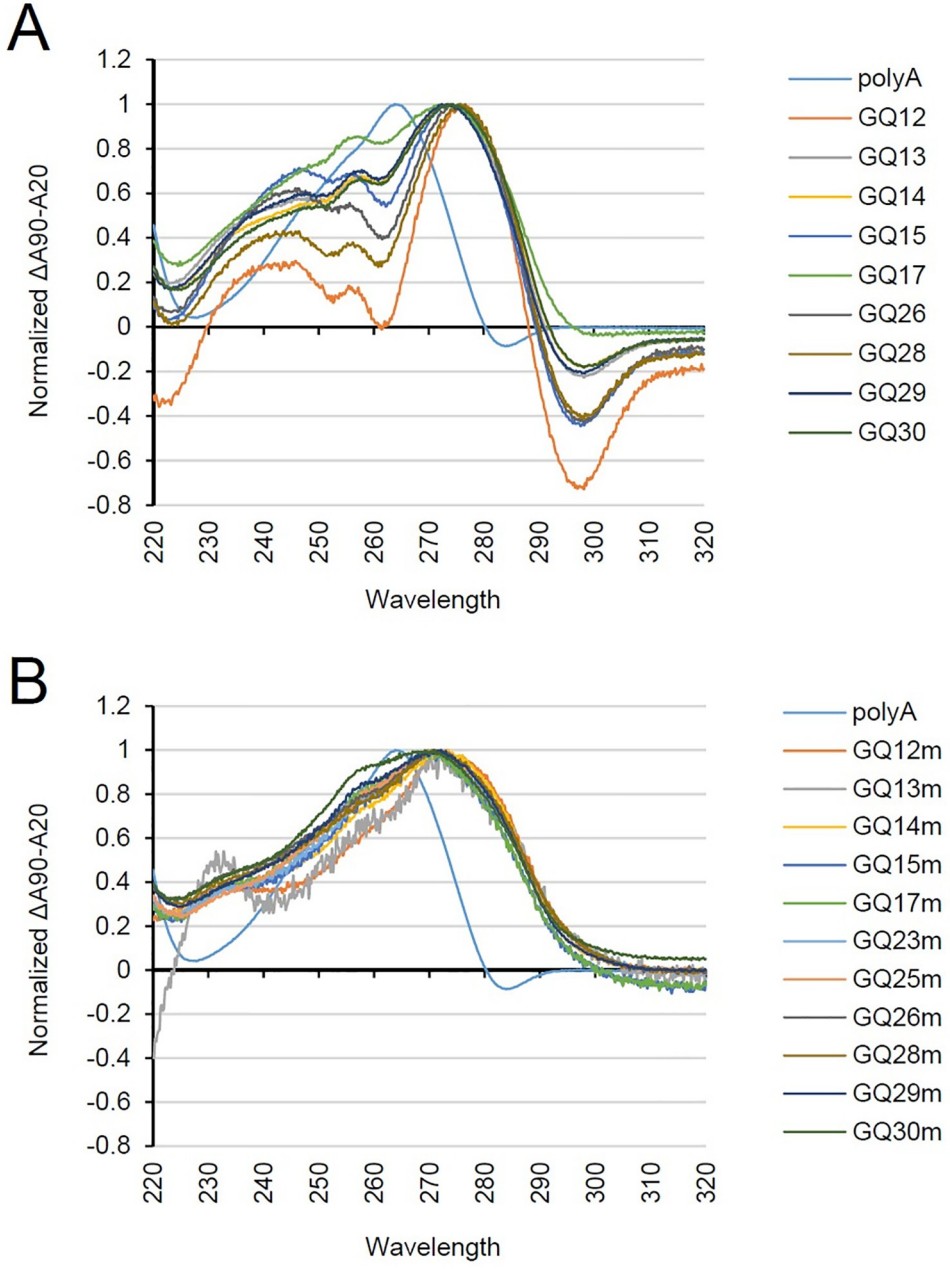

**Fig 4. Normalized TDS spectra of ORF14 GQ ODNs from pOka.** TDS spectra of wild-type (A) and mutant (B) ORF14 GQ ODNs (15 μM) from pOka annealed in 10 mM Tris-HCl pH 7.5 and 100 mM KCl are shown. Each spectrum was an average of three accumulations in the wavelength range between 220–320 nm. The spectra were blanked with buffer only before TDS calculation.

(S4 Fig). This ORF14-G4m DNA contains the same coding potential as the wild-type ORF14 (S5 Fig). The wild-type ORF14 and ORF14-G4m DNAs were cloned into the pSG5 vector containing the simian virus 40 early promoter [57] to express gC with a C-terminal HA-tag. gC is expressed as a ~64-kDa unmodified form and larger glycosylated forms [58]. In immunoblotting of lysates of 293T cells transfected with wild-type ORF14-HA and ORF14-G4m-HA plasmids, we observed the unmodified form of gC-HA largely accumulated in transfected 293T

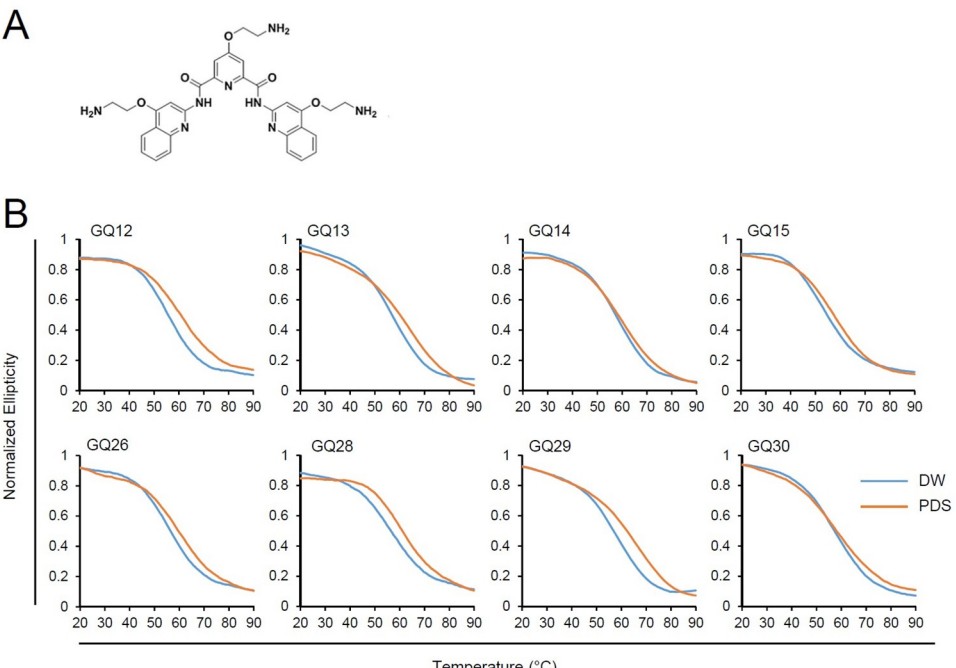

**Fig 5. Effect of PDS on thermal melting curves of ORF14 G4s.** (A) Chemical structure of PDS. (B) The CD melting curves of ORF14 GQ ODNs from pOka in the presence of DMSO and PDS are compared. GQ ODNs (15 μM) were annealed in the presence of 10 mM Tris-HCl (pH 7.5) and 100 mM KCl buffer with DMSO or 30 μM PDS (DNA to chemical ratio 1:2). The CD melting graph was calculated at 290 nm wavelength for all GQ ODNs. The melting curves obtained with mean values from triplicate results are shown. The data were normalized using the maximum ellipticity and smoothed using GraphPad Prism 5.

cells, although some modified forms (probably, glycosylated forms) were also detectable (Fig 7A). The unmodified form of gC-HA showed a slightly larger size than expected due to the addition of extra-amino acids between gC and the HA tag, which was introduced during cloning using LR clonase. We found that gC-HA expression was higher with the G4m plasmid than with the wild-type plasmid, suggesting that the G4 formation in ORF14 inhibits gC expression.

**Table 1. Effect of PDS on the Tm values of ORF14 G4s.**

| G4 motif | Tm (°C) ± PDS[a] | | |
|---|---|---|---|
| | DW[b] | PDS | ΔTm |
| GQ12 | 55.89±0.7 | 60.73±0.88 | 4.83 |
| GQ13 | 56.83±0.64 | 62.38±0.54 | 5.55 |
| GQ14 | 58.02±0.1 | 59.86±0.33 | 1.84 |
| GQ15 | 53.98±0.49 | 57.53±0.09 | 3.54 |
| GQ26 | 56.02±0.49 | 60.04±0.91 | 4.02 |
| GQ28 | 56.64±0.99 | 62.02±0.88 | 5.38 |
| GQ30 | 57.29±0.25 | 57.79±0.57 | 0.49 |

[a]Tm represents the mean and ±SD of 3 independent replicates.
[b]DW; distilled water

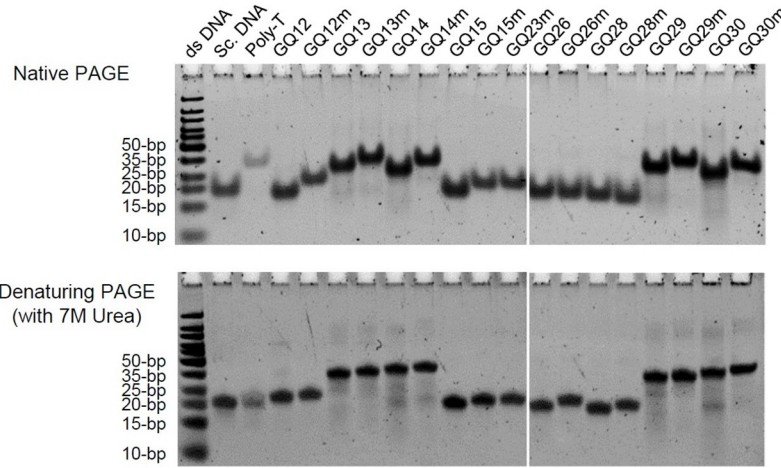

**Fig 6. Non-denaturing polyacrylamide gel electrophoresis (PAGE) analysis of GQ oligonucleotides.** Wild-type and mutant ORF14 GQ ODNs (1 μM) were subjected to native PAGE analyses after annealing in 100 mM K⁺ buffer (top) or denaturing PAGE with 7M urea (bottom). Scrambled (Sc DNA; 5′-TAACCGATGATATGAGTCAGATATAT-3′) and poly-T ODNs (Poly-T; 26mer) used as controls are shown. Double-stranded DNA (ds DNA) size makers are shown.

G4 formation on the template strand is known to inhibit gene transcription [59–61]. Therefore, we investigated whether the observed effect of G4 formation on gC expression occurs at the transcriptional level. We transfected 293T cells with wild-type and G4m ORF14 plasmids and treated cells with PDS, and gC mRNA levels were measured by qRT-PCR. The level of gC mRNA in PDS-untreated cells was higher with ORF14-G4m than that with wild-type ORF14, and the gC mRNA levels from wild-type ORF14 were dose-dependently reduced by PDS treatment, while those from ORF14-G4m were not affected (Fig 7B). These indicate that the ORF14 gene expression is suppressed by G4 formation at the transcriptional level.

To further support the G4-mediated suppression of gC expression, we constructed two more ORF14 mutant genes (G4m-1 and G4m-2). Compared to G4m, which contains all GQ mutant sequences (from GQ12m to GQ41m), G4m-1 contains only GQ34m to GQ41m sequences, whereas G4m-2 contains only GQ18m to GQ41m sequences. When the gC mRNA levels were compared among G4m, G4m-1, and G4m-2 ORF14 genes by qRT-PCR in transfected cells, both G4m-1 and G4m-2 also expressed higher levels of gC mRNA than wide-type, but the increase of gC mRNA by these mutants was much less than that by G4m (Fig 7C). Furthermore, PDS treatment repressed gC transcription from G4m-1 and G4m-2 but not from G4m (Fig 7C). The increased gC-HA mRNA transcription in the G4-disrupted ORF14 was not due to the enhanced stability of gC-HA mRNA since the stability of mRNAs produced from wild-type and G4m ORF14 was similar (Fig 7D). We observed that the difference in gC mRNA levels between wild-type and G4m was less than that in gC protein levels in transfected cells. Although the mutations introduced did not affect mRNA stability, it cannot be excluded that they also affected protein expression at the post-transcriptional level in transfection assays. The production of gC mRNA from the untagged ORF14 plasmid was similarly increased by G4-disrupting mutations, indicating that the HA tag did not affect ORF14 transcription (Fig 7E). Together, these results support the G4-mediated suppression of ORF14 expression and also demonstrate that the more G4 is formed, the more ORF14 is suppressed.

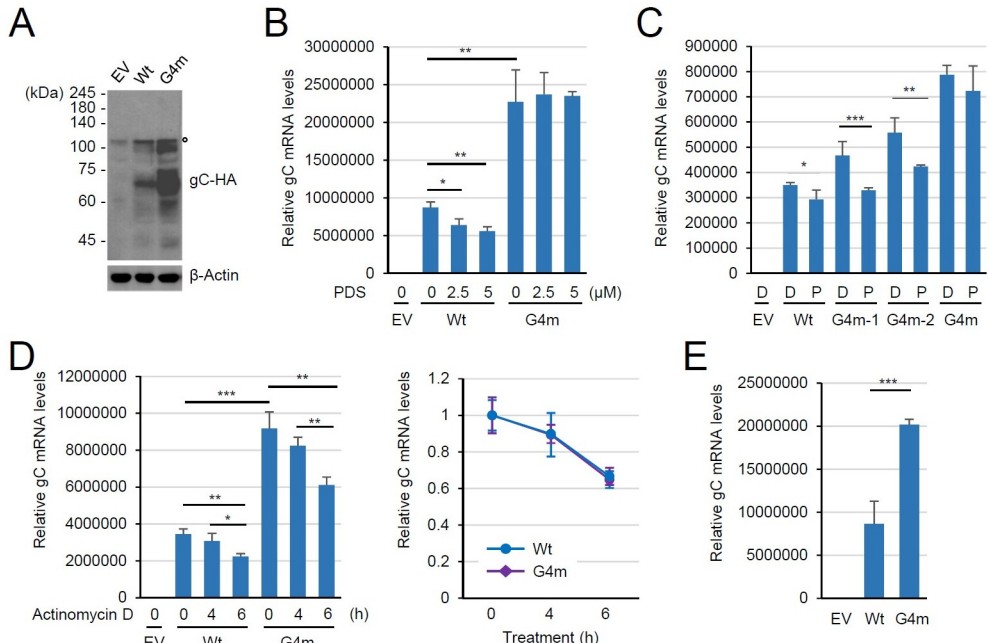

**Fig 7. Effect of G4 disruption in ORF14 on gC expression in transfected cells.** (A) 293T cells (1x10^6 cells per well) in six-well plates were transfected with 2 μg of empty vector (EV) or plasmids expressing pOka ORF14-HA or its G4-disrupted form (G4m). Total cell lysates were prepared in RIPA buffer at 72 h after transfection. The total cell lysates were subjected to immunoblotting with anti-HA antibody. Non-specific bands are indicated by the open circle. The levels of β-actin are shown as a loading control. (B) 293T cells were transfected as in (A) and treated with DW or increasing amounts (2.5 and 5 μM) of PDS at 16 h after transfection. At 72 h after transfection, total RNA was prepared and the ORF14 mRNA levels were measured by qRT-PCR and normalized with those of β-actin. Results are shown as mean values and standard errors of triplicates. (C) 293T cells (1x10^6 cells per well) in six-well plates were transfected with 2 μg of empty vector (EV) or plasmids expressing wild-type ORF14-HA (Wt) or its different G4-mutated forms (G4m-1, G4m-2, or G4m) and treated with DW (D) or 2.5 μM of PDS (P) at 16 h after transfection. At 72 h after transfection, total RNA was prepared and the ORF14 mRNA levels were measured by qRT-PCR as in (B). (D) 293T cells were transfected as in (A) and treated with actinomycin D for the indicated times. The gC mRNA levels were determined as in (B and C). (E) 293T cells were transfected with 2 μg of empty vector (EV) or plasmids expressing pOka ORF14 (Wt) or its G4-disrupted form (G4m). The gC mRNA levels were determined as in (B) and (C).

## G4 formation suppresses gC expression at the transcriptional level during virus infection

We further investigated the suppressive effect of G4 formation on gC expression using recombinant viruses. We first produced a recombinant pOka ORF14-HA virus that expresses C-terminal HA-tagged gC using bacmid mutagenesis (S6A Fig). This recombinant virus is designed to express gC-HA without inserting additional amino acids between gC and HA. The pOka bacmid contains a green fluorescent protein (GFP) expression cassette between ORF60 and ORF61 and a luciferase reporter gene driven by the simian 40 virus promoter between ORF66 and ORF67 [37]. The recombinant viruses were grown in MeWo cells after transfection with bacmid DNAs. Human fibroblast (HF) cells were infected with wild-type pOka (prepared from the pOka bacmid) and pOka-ORF14-HA viruses and their growth was measured by luciferase activity. The growth of the ORF14-HA virus was similar to that of the wild-type virus, indicating that the C-terminal HA tag on gC did not affect viral replication (Fig 8A). In immunoblotting with anti-HA antibody, the gC-HA proteins in ORF14-HA virus-infected cells were expressed as highly glycosylated forms with a major form between 100 and 140 kDa (Fig 8B).

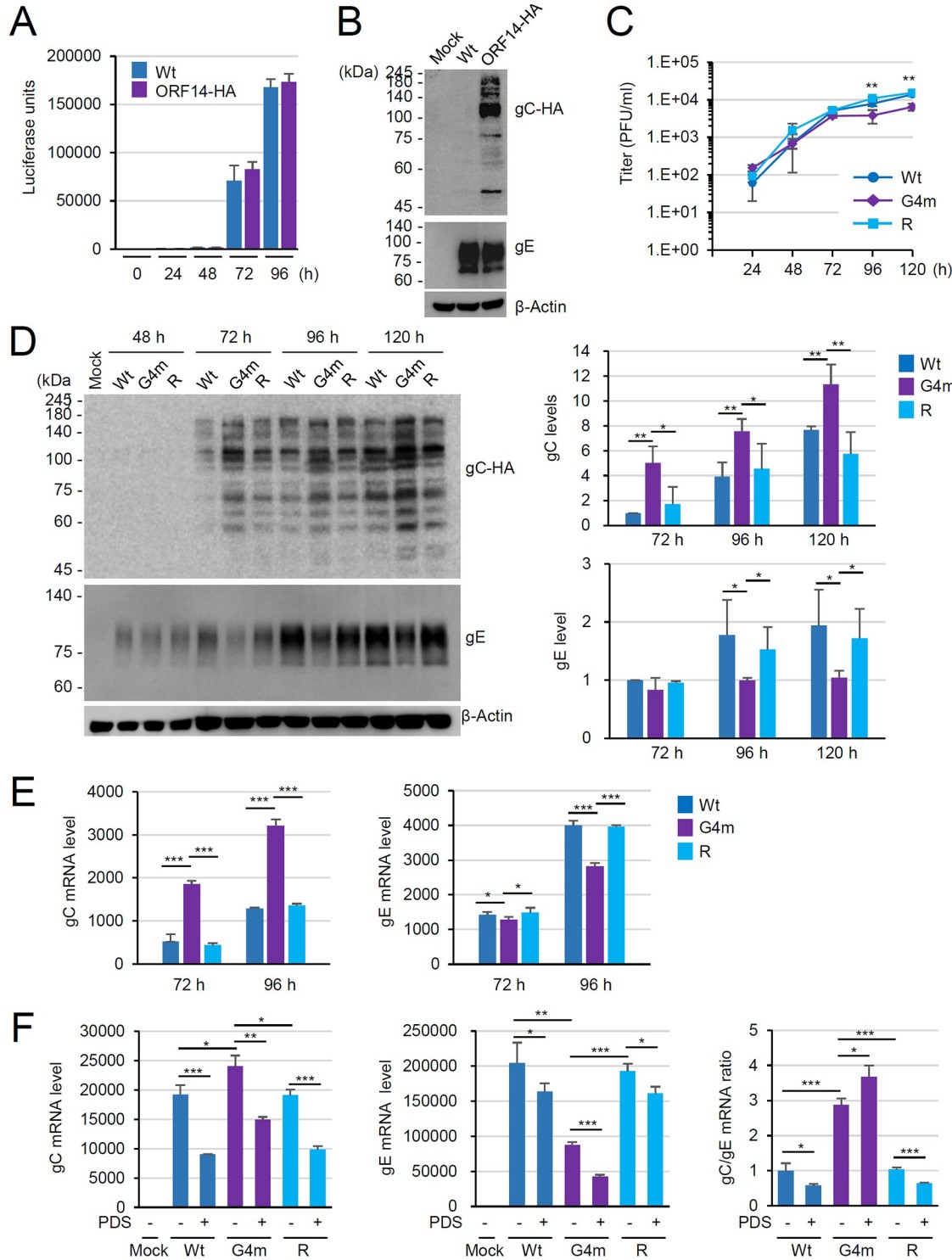

**Fig 8. Altered gC expression in the recombinant virus containing the G4-disrupted ORF14-HA gene.** (A) HF cells were infected with recombinant pOka viruses containing the wild-type ORF14 or ORF14-HA gene at an MOI of 0.01. Cells were harvested at 24, 48, 72 and 96 h after infection and luciferase activities in cell lysates were measured. Results are shown as mean values and standard errors of three independent experiments. (B) HF cells were infected with recombinant viruses (wild-type or ORF14-HA) at an MOI of 0.01 Total cell lysates were prepared at 96 h after infection and subjected to SDS-PAGE and immunoblotting with ant-HA and anti-gE (ORF68) antibodies. Levels of β-actin are shown as a loading control. (C) HF cells in 12-well plates were infected with recombinant pOka viruses [wild-type, G4-disrupted mutant (G4m), and its revertant (R)] at an MOI of 0.001. Cells were harvested at

24, 48, 72, 96, and 120 h after infection. The cells were trypsinized and the virus titers were determined by plaque assay. Results are shown as mean values and standard errors of three independent experiments. p-values <0.01 (**) are indicated when significance was observed both between G4m and Wt and between G4m and R. (D and E) HF cells in 12-well plates were infected with recombinant viruses as in (C). Total cell lysates were also prepared at 48, 72, 96, and 120 h after infection and the expression levels of gC-HA and gE were determined by immunoblotting with anti-HA and anti-gE antibodies, respectively. Levels of β-actin are shown as a loading control (panel D, left). The gC-HA and gE bands in three independent experiment were quantitated with ImageJ and shown as graphs (panel D, right). Total mRNA was also prepared at 72 and 96 h after infection and the mRNA levels of gC (ORF14) and gE (ORF68) were determined by qRT-PCR. Results are shown as mean values of viral mRNA normalized with β-actin and standard errors of three independent experiments (E). (F) HF cells in 12-well plates were infected with recombinant viruses as in (C). At 72 h after infection, cells were treated with PDS (5 μM) for 48 h before qRT-PCR assays. The gC and gE mRNA levels and the gC/gE ratio are shown as in (E).

We then produced the recombinant virus containing the G4 formation-defective ORF14 gene. The G4 motifs-containing region in the pOka ORF14-HA bacmid was replaced with the corresponding region from the ORF14-G4m plasmid, resulting in the ORF14-G4m-HA bacmid. Its revertant bacmid, ORF14-G4m-HA-(R), was also produced from the ORF14-G4m-HA bacmid (S6B Fig). Recombinant viruses were grown in MeWo cells transfected with the bacmid DNAs. HF cells were then infected with recombinant viruses [ORF14-HA, ORF14-G4m-HA, or ORF14-G4m-HA-(R)] at an MOI of 0.001 and their growth was compared by determining cell-associated virus titers by performing plaque assays in MeWo cells. The ORF14-G4m-HA virus showed slightly reduced growth compared with that of the wild-type and revertant viruses (Fig 8C).

We next compared the gC levels in HF cells infected with recombinant viruses at an MOI of 0.001 by immunoblot analysis. During ORF14-G4m-HA virus infection, gC was detected as early as 72 h at a significantly higher level than those in wild-type and revertant viruses (Fig 8D). As a control, gE (encoded by ORF68) was detected from 48 h during ORF14-G4m-HA virus infection and its levels were comparable at 48 h and lower at 96 and 120 h compared to those in wild-type and revertant viruses (Fig 8D). The delayed expression of gC compared with that of gE during wild-type virus infection was consistent with the previous reports [62–64]. We also compared the gC and gE mRNA levels in recombinant virus-infected HF cells using qRT-PCR. The gC mRNA levels in G4m virus were much higher at 72 h and 96 h than those in wild-type and revertant viruses (Fig 8E, left), while gE mRNA levels in G4m virus similar or reduced compared to those in wild-type and revertant viruses (Fig 8E, right), demonstrating that G4 formation in ORF14 indeed inhibits gC expression at the transcriptional level during virus infection. We also investigated the effect of PDS on gC transcription in virus-infected cells. Since the G4 ligands could affect the expression of several viral and cellular genes from the early stage of infection, PDS treatment reduced gC and gE transcription from both wild-type and G4-disrupted (G4m) viruses (Fig 8F, left and center); however, the gC/gE mRNA ratio was higher by 3-fold in G4m virus than in wild-type and revertant viruses (Fig 8F, right). Furthermore, the gC/gE mRNA ratio was decreased by PDS treatment in wild-type and revertant viruses, while it was increased in G4m virus (Fig 8F, right). These results demonstrate that gC transcription was less severely suppressed by PDS during G4m virus infection than in wild-type virus infection.

## Enhanced gC expression in the ORF14-G4m virus attenuates viral cell-to-cell spread

Previous studies suggested that enhanced gC expression may result in the reduction of viral cell-to-cell spread in VZV and Marek's disease virus (MDV), an alphaherpesvirus of chickens [62,65]. Therefore, we also investigated whether the ORF14-G4m virus displays a similar growth property. MeWo cells were transfected with the VZV bacmids containing the

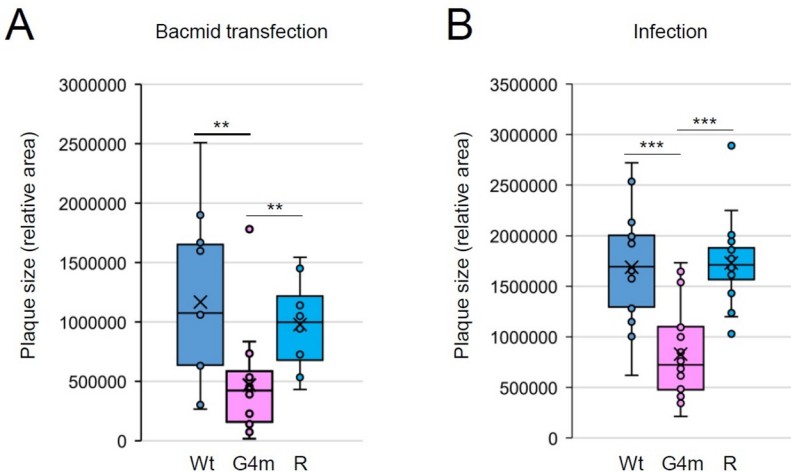

**Fig 9. Comparison of plaque sizes of ORF14 recombinant viruses.** (A) MeWo cells in six-well plates were transfected with the VZV bacmids (1 μg) containing the ORF14-HA gene [wild-type, G4m, or revertant (R)]. At 8 days after transfection, images of GFP+ plaques were taken. A similar plaque size reduction pattern was found with the G4m virus in two independent experiments. Since the efficiency of plaque generation in bacmid-transfected cell is low, we combined all plaques (n = 11 for Wt, n = 17 for G4m, and n = 8 for G4m-R) in two experiments to make the graph. Plaque area size was measured with ImageJ and shown as a box and whisker graph. Box plots show the first and third quartiles as a box. Horizontal lines are samples median. X corresponds to the mean. Whiskers correspond to 1.5 times the inter-quartile distance (IQR) from the difference between first and third quartiles. Images showing plaques are shown in S7A Fig. (B) MeWo cells in six-well plates were infected with cell-associated recombinant viruses. At 5 days after infection, cells were stained with crystal violet and plaque size was measured (n = 20) and shown as a graph as in (A). The experiments were performed more than three times and showed similar results. The data from a representative experiment are shown. Plaques images are shown in S7B Fig.

ORF14-HA, ORF14-G4m-HA, or its revertant gene and GFP+ plaque size was measured at 8 days after transfection. The plaque size of the ORF14-G4m virus was significantly smaller than those of the wild-type and revertant viruses (Figs 9A and S7A). The similar small plaque property of the ORF14-G4m virus was observed when MeWo cells were infected with cell-associated recombinant viruses (Figs 9B and S7B). A similar small plaque phenotype of the ORF14-G4m virus was observed in HF cells (S8 Fig). These results demonstrate that the enhanced gC expression in the ORF14-G4m virus is inhibitory for viral cell-to-cell spread.

## Discussion

Our genome-wide analysis of G4 motifs in the VZV genome revealed that they are enriched in the $IR_S$ and the $TR_S$ regions and in some of the R regions (R2, R3, R4A, and R4B). Each R region has its unique repeat sequences (elements) and the copy number of elements varies among the different VZV strains [26]. The G4 motifs found in R2 have interesting features. First, nucleotide variations found in R2 mostly occur in relation to G4 motifs. Comparative analysis of nine glycoprotein genes in different strains indicated a higher level of single nucleotide polymorphisms in ORF14 [64]. It is intriguing to speculate that G4 formation during gene transcription or DNA replication is responsible for this high level variation in ORF14. Second, the G4 motifs are located at the junction between two elements, suggesting the involvement of G4s in determining the copy numbers of elements. G4s may cause RNA polymerase pausing and DNA nicks, promoting recombination. Notably, in the absence of BLM helicase, of which mutation can cause Bloom syndrome, a cancer predisposition disorder, sister chromatid exchange events are enriched at sites with G4 motifs in transcribed genes [66]. Furthermore, the recombination breakpoints found in the HSV-1 genome are associated with G4 motifs,

suggesting a role of G4s in recombination during HSV-1 infection [67]. The numbers of repeated units in R2 vary in strains, although they seem to be relatively stable during viral replication [68]. Whether the copy numbers of elements in the R regions affect the VZV life cycle is not clear. However, it is conceivable that recombination at the R regions maintains the genome length and G4 formation is involved in this process by promoting recombination.

In this study, we provided biophysical evidences that several G4 motifs found in R2 within ORF14 form a G4 structure. Analyses of ODNs showed that nine motifs out of 11 predicted motifs, except GQ21 and GQ22, formed an anti-parallel G4s in CD. Among these nine G4 motifs, eight G4 motifs such as GQ12, 13, 14, 15, 26, 28, 29, and 30, except GQ17, showed the G4 characteristic in TDS. Therefore, we considered that these eight G4 motifs can form a G4 structure. In the thermostability assays using PDS, the stability of G4s formed by these G4 motifs could be increased by PDS. The native PAGE analysis demonstrated that GQ12, 13, 14, 15, 29, and 30 can form a compact intramolecular structure. These sequences for these six G4 motifs are distributed throughout the ORF14 R2, indicating that G4 formation is a prominent structural characteristic in the R2 region.

We showed that the G4 formation in R2 (on the template strand of ORF14) negatively regulates gC transcription. This finding is consistent with the notion that G4 formation on the template strand can inhibit transcriptional elongation by RNA polymerases. VZV expresses 11 glycoproteins that play important roles in virus attachment, membrane fusion, and pathogenesis [69]. Unlike gC proteins in HSV-1 and PRV [70,71], gC in VZV is not required for virion binding to cell surface receptors for virus entry, since the mutant viruses that lack gC expression normally grow in cell culture [27,72]. However, gC appears to be involved in tissue tropism, since the gC-defective mutant viruses show poor growth in human skin tissue [27,28]. Previous studies showed that expression level of gC varies among strains, with a lower level of expression in pOka than Scott [58]. gC is a true late protein, as its transcription occurs at very late during VZV infection probably due to the lack of the factors required for gC synthesis during early phases of infection [62–64]. We also observed delayed expression of gC compared with that of gE. Furthermore, our analysis of the G4 formation in R2 revealed that gC transcription is suppressed by G4s. Whether delayed expression of gC is solely attributed to the G4-mediated suppression of gC transcription is not clear. However, our results provide a clue for this long-standing question of gC expression.

We also found that the enhanced gC expression in the mutant virus containing the G4-disrupted ORF14 gene led to a small plaque phenotype. Previous studies reported that higher gC expression of VZV or MDV led to a reduction of plaque size [62,65]. We also verified that higher gC expression by the ORF14-G4m virus results in a decreased plaque size in both bacmid DNA-transfected or recombinant virus-infected cells. The ORF14-G4m virus also showed a slightly reduced growth. This is consistent with findings in MDV showing that the overexpression of gC reduced viral growth [62,65]. Our results support the notion that higher expression of gC has a negative impact on the cytopathology and viral growth rate during VZV growth. Since the copy number varies in some strains, it is speculated that strains with more 42-bp elements in ORF14 will express less gC due to G4 formation and show better cell-to-cell spread.

González-Motos et al. recently showed that gC of VZV binds to chemokines with high affinity and facilitates the migration of leukocytes [73]. VZV infects dendritic cells and T lymphocytes, and this may be important for systemic dissemination of the virus [74,75]. Since gC is expressed at true late stages of the virus life cycle [63,76], the delayed gC expression by G4s may be a viral strategy to optimize the timing of virus dissemination, unless the virus is eradicated by immune responses before maturation processes for infectious virions. Further studies

are necessary to elucidate how G4-mediated regulation of gC expression affects VZV replication *in vivo*.

# Materials and methods

## Cells and chemicals

Immortalized human melanoma MeWo cells, human embryonic kidney (HEK) 293T cells, and human foreskin fibroblast (HF) cells were grown in Dulbecco's modified Eagle medium (GIBCO) supplemented with 10% fetal bovine serum and 100 U of penicillin-streptomycin at 37°C and 5% $CO_2$. PDS was purchased from Cayman.

## *In silico* G4 prediction

The Dumas VZV genome (NC_001348.1) and pOka VZV genome (AB097933.1) were mined for potential non-overlapping G4 motifs using the Quadparser program [77]. We divided the G4s into conventional, long-loop, and bulged categories based on the schema used for prediction of the G4s. Conventional G4s were predicted using the schema $G_{3-6}N_{1-7}G_{3-6}N_{1-7}G_{3-6}N_{1-7}G_{3-6}$ and represent the conventional 3-stack G4s. In our case, long-loop G4s were restricted to a single long loop and predicted using the schema $G_{3-6}N_{8-30}G_{3-6}N_{1-7}G_{3-6}N_{1-7}G_{3-6}$ (and subsequent loops). Bulged G4s were predicted using the schema $G_{3-6}N_{1-7}G_{3-6}N_{1-7}G_{3-6}N_{1-7}G_2(A/T/C)G$ and remaining loop permutations. Duplicates were removed if the same sequence was predicted under the same category (long-loop, conventional, or bulged). Sequences were retained if predicted under different categories.

## Circular dichroism (CD) spectroscopy and thermal melting curve analysis

ODNs used for the CD spectroscopy are described in S1 Table. CD spectroscopy was performed on a Jasco J-810 spectroscopy fitted with a Peltier temperature controller (CDF426S, Jasco, Japan). The ODNs were dissolved at a concentration of 15 μM in a buffer containing 10 mM Tris-HCl (pH 7.5) and 100 mM KCl, followed by denaturation at 95°C for 5 min and annealing at room temperature for 2 h. For studies with G4-binding ligands, preformed G4s were treated with 30 μM PDS for a DNA-to-ligand ratio of 1:2. CD spectra were measured at 25°C using a 1 mm path length quartz cuvette (Hellma, Germany) as the average of three accumulations between 230–320 nm, with a response time of 2 sec, scanning speed of 100 nm/min, and data pitch of 1 nm. CD melting curves were recorded between 20–90°C at a wavelength of 290 nm for all nucleotides with 2°C/min heating rate. After subtracting the spectrum of buffer only from all samples, the data were normalized to the maximum ellipticity. The first derivative of the melting curve was plotted and fitted and smoothed using inbuilt functions in GraphPad Prism 5.

## Thermal Difference Spectra (TDS)

The UV spectra was measured on a Jasco V-750 UV spectrophotometer fitted with a Peltier temperature controller (ETCS-761, Jasco, Japan). The ODNs were dissolved at a concentration of 15 μM in a buffer containing 10 mM Tris-HCl (pH 7.5) and 100 mM KCl, followed by denaturation at 95°C for 5 min and annealing at room temperature for 2 h. The UV spectra was measured using a 1 mm path length quartz cuvette (Hellma, Germany) at 20°C and 90°C following which the spectra were blanked using buffer only spectra. Finally, the TDS was measured as the difference between absorbance at 20°C and 90°C. The data was normalized to the maximum TDS value to obtain the normalized TDS.

## Plasmids

The ORF14 of pOka was amplified by PCR as a SalI-NotI fragment from the pOKa bacmid with specific primers (LMV2825/2847 for HA-tagged plasmid or LMV2825/2826 for untagged plasmid) (S2 Table). The PCR products were cloned into the SalI and NotI sites of the pENTR-1A vector (Invitrogen). The G4-defective ORF14 (G4m) DNA was chemically synthesized (Synbio Technologies and Cosmogentech Inc.) and cloned into HindIII-BamH1 sites in pUC57. The G4-defective ORF14 was then PCR amplified with primers (LMV2825/2847) and cloned into the SalI and NotI sites of pENTR-1A. Expression plasmids encoding C-terminal HA-tagged ORF14 were generated by transferring the DNA in pENTR-1A to the pSG5-based destination vector using LR clonases (Invitrogen).

## Antibodies

Anti-HA mouse monoclonal antibodies (MAbs) 3F10 conjugated with horseradish peroxidase (HRP) and 12CA5 were purchased from Roche. Anti-gE mouse MAb 9C8 was purchased from Santa Cruz Biotechnology. Anti-β-actin mouse MAb GT5512 was purchased from Gene Tex, Inc.

## Immunoblot analysis

Cells were washed with PBS and total cell lysates were prepared in RIPA buffer. Equal amounts of clarified cell extracts were separated on an SDS-polyacrylamide gel and electroblotted onto nitrocellulose membranes. Blots were blocked with PBS plus 0.1% Tween 20 (PBST) containing 5% nonfat dry milk for 1 h at room temperature. After three washes with PBST, blots were incubated with the appropriate antibodies in PBST for 1 h at room temperature or overnight at 4°C. After three 5-min washes with PBST, blots were incubated with HRP-conjugated goat anti-mouse IgG secondary antibody (Amersham) for 1 h at room temperature. For HA-tagged proteins, HRP-conjugated anti-HA antibody was used. Blots were then washed three times with PBST and the protein bands were visualized with an enhanced chemiluminescence system (Amersham).

## Bacmid mutagenesis

The pOka bacmid [37], which was kindly provided by Dr. Hua Zhu (Rutgers-New Jersey Medical School, NJ, USA), was used as a template for mutagenesis. The pOka bacmid containing the ORF14-HA gene was generated using a counter-selection bacterial artificial chromosome (BAC) modification kit (Gene Bridges, Germany). Briefly, the *rpsL-neo* cassettes flanked by a 50-nucleotide homology arm targeting the C-terminal region of ORF14 were PCR amplified (with LMV2877 and LMV2878) and introduced into *E. coli* DH10B containing wild-type pOka bacmid for recombination by electroporation using a Gene Pulser II (Bio-Rad). The primer sequences used for ORF14 bacmid mutagenesis are shown in S2 Table. The *rpsL-neo* containing intermediate bacmid was selected on Luria Broth (LB) agar plates containing kanamycin. In second round of homologous recombination, the *rpsL-neo* marker cassette was replaced by the annealed 130 nucleotide homologous arms containing an HA-tag (LMV2879 and LMV2880), resulting in recombinant bacmid containing the ORF14-HA gene. The mutation in the bacmid was confirmed by direct sequencing.

The G4-disrupted mutant (G4m) ORF14 gene was generated in the ORF14-HA bacmid background. The *rpsL-neo cassettes* flanked by 50 nucleotide homology arms upstream and downstream of the target region (i.e., the G4-forming region) were PCR amplified using specific primer sets (LMV2881 and LMV2882). The amplified *rpsL-neo* fragments were purified

and introduced via electroporation into *E. coli* DH10B containing the G4m sequences for recombination. The intermediate pOka bacmid construct containing the *rpsL-neo* cassette was selected on LB agar plates containing kanamycin. Next, to replace the *rpsL-neo* cassette, the G4-disrupted ORF14 DNA fragments were PCR amplified with specific primers (LMV2883 and LMV2884) and were recombined into the pOka bacmid containing the *rpsL-neo* cassette. *E. coli* cells containing the pOka-ORF14-G4m bacmid were selected on LB plates containing streptomycin. Mutated regions were amplified and sequenced to verify the desired mutations. To generate the revertant pOka bacmid from ORF14-G4m, the intermediate pOka bacmid construct containing the *rpsL-neo* cassette was selected and the *rpsL-neo* cassette was replaced with DNA fragments containing the wild-type ORF14 sequences, which were PCR amplified with specific primers (LMV2883 and LMV2884), by homologous recombination as described above.

To generate recombinant viruses, pOka bacmids were introduced into MeWo cells by transfection using the OmicsFect reagents (Omics Bio) according to the manufacturer's protocol. After 10 days post-transfection, the cells were transferred to new MeWo cell culture to propagate the viruses. The titers of propagated virus stocks were determined using plaque assay with MeWo cells.

## Plaque assays

Plaque assays were performed to determine titers of recombinant viruses. MeWo cells were infected with serial 1 to 10 dilutions of viruses. At 5 days post-infection, the medium was discarded, and cells were stained with crystal violet for 6 h prior to plaque counting.

## Quantitation of viral mRNAs

Total RNA was extracted from virus-infected cells ($5 \times 10^5$) using TRI reagents (Molecular Research Center). RNA quality was checked by determining the integrity of rRNAs in agarose gel electrophoresis. RNA to cDNA EcoDry Premix (TaKaRa) was used to generate cDNAs. qRT-PCR was performed using the Power SYBR Green PCR Master Mix and QuantStudio Real-Time PCR System. The primers used for ORF14 and ORF68 were 5′-GGATGCA TAGGGGTTGCGATAA-3′ (ORF14 forward), 5′-TGCATCTACCTACGCCACTA-3′ (ORF14 reverse), 5′-GTACATTTGGAACATGCGCG-3′ (ORF68 forward), and 5′-TCCACATAT GAAACTCAGCCC-3′ (ORF68 reverse) [76]. The primers for β-actin were 5′-AGCGG GAAATCGTGCGTG-3′ (forward), and 5′-CAGGGTACAT GGTGGTGCC-3′ (reverse).

## Luciferase reporter assays

Cells were collected and lysed by three freeze-thaw steps in 100 μl of 0.25 M Tris-HCl (pH 7.9) with 1 mM dithiothreitol. Cells extracts were clarified in a centrifuge and 20 μl of extracts were incubated with 350 μl of reaction buffer A (25 mM glycylglycine [pH 7.8], 15 mM $MgSO_4$, 5 mM ATP, and 4 mM EGTA) and then mixed with 100 μl of 0.25 mM luciferin (Sigma-Aldrich) in reaction buffer A. The TD-20/20 luminometer (Turner Designs) was used for a 10-s assay of the photons produced (measured in relative light units).

## Statistical analysis

Statistical significance are determined using Student's *t*-test and indicated by p-values $<0.05$ (*), $<0.01$ (**), and $<0.001$ (***) are indicated.

## Supporting information

**S1 Appendix. Putative G4 motifs predicted in the VZV genomes (Dumas and pOka).**
(XLSX)

**S2 Appendix. Nucleotide variations found in the R2 reiteration sequence regions of VZV.**
(XLSX)

**S1 Fig. Putative G4 motifs in IR$_S$.** (A) The positions of ORFs and OriS in the IR$_L$, IR$_S$, U$_S$ and
TR$_S$ regions in the VZV genome are shown. (B) The G4 motifs (GQ64 to GQ104) found in IR$_L$
and IR$_S$ are indicated on the top and bottom strands of the genome. Similar G4 motifs with
overlapping sequences are clustered in a box. The types of G4 (conventional, long-loop, and
bulged) are indicated with different colors.
(TIF)

**S2 Fig. Indication of putative G4 motifs in IR$_S$ on the VZV genome sequence.** The G4
motifs (GQ65 to GQ104) found in IR$_S$ are indicated below the genome sequence of Dumas.
The G4 motifs predicted for conventional, long-loop, and bulged G4s are indicated as blue,
black, and red lines, respectively. The positions of ORF61, ORF62, OriS, and ORF63 are indi-
cated. In OriS, the locations of the R4A reiteration sequence, Boxes A, B, and C, and palin-
drome sequence are indicated.
(TIF)

**S3 Fig. Conservation of G-runs and nucleotide variations in R2 reiteration sequences.**
Multiple nucleotide sequence alignment of the R2 reiteration regions in ORF14 of 141 VZV
strains was performed using MAFFT [78,79]. R2 of the DUMAS strain (from 20,681 to
21,017 in the genome) was used as the reference sequence. Graphical representation of R2
variation indicates preserved G-runs (3 Gs) in orange, other preserved sequences in gray, and
sequences with variations in black letters. Nucleotide sequences with <1% variations are rep-
resented by a single black letter. The degree of variation in nucleotides is indicated as a per-
centage.
(TIF)

**S4 Fig. Comparison of wild-type and G4-disrupted mutant (G4m) ORF14 gene
sequences.** Alignment of the top strand sequences for part of ORF14 from 20602 to 21033
(in pOka) with its corresponding G4-disrupted mutant (G4m) sequences is shown. The
identical sequences are denoted as *. The positions of G4 motifs (GQ12 to GQ41) in pOka
are indicated below the sequences. Black arrows, long-looped G4; red arrows, bulged G4.
The sequences corresponding to the translation initiation codon for gC are indicated in the
box.
(TIF)

**S5 Fig. Complete sequences of the G4-disrupted ORF14 gene and its translation product.**
(A) The complete sequences on the template strand of the wild-type ORF14 gene of pOka and
its G4-disrupted mutant (G4m) are shown. The mutated sequences in G4m are highlighted.
(B) The gC amino acid sequences produced from the wild-type and G4m ORF14 genes are
compared and show 100% identity. The repeated sequences of 14 amino acids encoded from
the R2 region are indicated as arrows.
(TIF)

**S6 Fig. Schemes for the production of recombinant viruses containing the ORF14-HA and
ORF14-G4m-HA gene.** The scheme for the production of pOka bacmid containing the

ORF14-HA gene (A) and pOka bacmids containing the ORF14-G4m-HA gene and its revertant gene (B). See the Materials and Methods for the detailed procedure.
(TIF)

**S7 Fig. Images showing plaque sizes of recombinant viruses in MeWo cells.** Images of MeWo cells used for measuring plaque sizes in Fig 9 are shown. The boundary of green fluorescence was used as a plaque boundary for measuring plaque regions in VZV-bacmid infected MeWo cells. (A) GFP images of cells showing plaques taken at 8 days after bacmid transfection. (B) Cell images showing plaques taken at 5 days after virus infection. To determine the plaque region in VZV-infected MeWo cells, empty regions for the confluent cell monolayer are measured.
(TIF)

**S8 Fig. Comparison of plaque sizes of ORF14 recombinant viruses in HF cells.** (A) HF cells in six-well plates were infected with cell-associated recombinant viruses. At 14 days after infection, cells were stained with crystal violet and plaque size was measured (n = 18). Plaque area size was measured with ImageJ in triplicate experiments and shown as a box and whisker graph. p-values <0.001 (***) are indicated. (B) Plaques images are shown as in S7 Fig.
(TIF)

**S1 Table. G4 motifs found in pOka R2 and their mutant sequences used in this study.**
(XLSX)

**S2 Table. Primers used for ORF14 cloning and bacmid mutagenesis.**
(XLSX)

## Acknowledgments

We thank Dr. Hua Zhu for providing the pOka bacmid.

## Author Contributions

**Conceptualization:** Woo-Chang Chung, Subramaniyam Ravichandran, Moon Jung Song, Kyeong Kyu Kim, Jin-Hyun Ahn.

**Formal analysis:** Woo-Chang Chung, Subramaniyam Ravichandran.

**Funding acquisition:** Jin-Hyun Ahn.

**Investigation:** Woo-Chang Chung, Subramaniyam Ravichandran, Daegyu Park, Gwang Myeong Lee, Young-Eui Kim, Youngju Choi.

**Supervision:** Jin-Hyun Ahn.

**Writing – original draft:** Woo-Chang Chung, Subramaniyam Ravichandran.

**Writing – review & editing:** Jin-Hyun Ahn.

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
