## [Decision Letter · Decision Letter 0]

11 Oct 2022

Dear Dr. Ahn,

Thank you very much for submitting your manuscript "G-quadruplexes formed by Varicella-Zoster virus reiteration sequences suppress expression of glycoprotein C and regulate viral cell-to-cell spread" for consideration at PLOS Pathogens. As with all papers reviewed by the journal, your manuscript was reviewed by members of the editorial board and by several independent reviewers. In light of the reviews (below this email), we would like to invite the resubmission of a significantly-revised version that takes into account the reviewers' comments.

There is a major concern from one reviewer (#2) that the observed phenotype may be the result of the HA tag on gC interrupting proper processing, as opposed to the mutations in the G4 quadruplex. This issue, as well as the multiple comments from reviewer #1, need to be addressed.

We cannot make any decision about publication until we have seen the revised manuscript and your response to the reviewers' comments. Your revised manuscript is also likely to be sent to reviewers for further evaluation.

Sincerely,

Robert F. Kalejta

Associate Editor

PLOS Pathogens

Blossom Damania

Section Editor

PLOS Pathogens

Kasturi Haldar

Editor-in-Chief

PLOS Pathogens

orcid.org/0000-0001-5065-158X

Michael Malim

Editor-in-Chief

PLOS Pathogens

orcid.org/0000-0002-7699-2064

There is a major concern from one reviewer (#2) that the observed phenotype may be the result of the HA tag on gC interrupting proper processing, as opposed to the mutations in the G4 quadruplex. This issue, as well as the multiple comments from reviewer #1, need to be addressed.

Reviewer's Responses to Questions

**Part I - Summary**

Reviewer #1: Chung and co-authors submitted a manuscript dealing with the presence of G-quadruplexes (G4s) in the genome of Varicella Zoster virus (VZV). In particular, the research work intends to shed light on the downregulating role of a cluster of repeated short regions, able to fold into G4s, on the expression of the viral glycoprotein C.

The work is clearly written, and the topic is very interesting. Nonetheless, there are issues in the experimental design and in the described results that need to be clarified.

Reviewer #2: In this manuscript, Chung et al report evidence to suggest that there high G/C regions of VZV include structures called G Quadruplexes. G Quad elements are known to form in single stranded DNA and RNA under some considtions to form different structures that can affect expression of RNA at the transcriptional level. They particularly focus on the VZV gC protein, which in most strains has 7/2/3 or a 42bp high G+C repeating element as part of its coding region. The work claims there is the formation of G Quadruplexes by the repeat element that regulates the transcription of gC so that, when such elements are mutated, there is a higher level of gC expression.

The work first involves scanning the genome for G Quadruplex formation elements using a program. They report the finding of about 150 potential elements. Including several found in the gC coding repeating elements. The then do a series of biophysical studies on the specific elements assessed outside of the context of their VZV environment using oligonucleotides and mutant oligonucleotides. They show subtle differences in circular Dichroism spectroscopy, Thermal difference spectra and the consequences of a known binding drug to G4 Quad pyridostatin. Which inhibits the actions ( it is not clear why the drug also affects the mutant oligonucleotides despite the text indicating it does not, See Figure 7b. Whiles this shows effects on the G4 Q oligonucleotides, it is of course separated from the context of the virus, so the intepretation of these data and if it occurs in the context of the virus is difficult. While interesting, I have several major concerns. I do not think this work is of sufficient significance and asdvancement for the field for publication in the Journal Plos Pathogens

Reviewer #3: The manuscript by Chung et al explores the potential role of G-quadruplexes for regulating expression of glycoprotein C in Varicella Zoster virus. This is a nicely presented story, beginning with a bioinformatic evaluation of potential G quadruplexes within the VZV genome and then biophysical analyses of G-quadruplexes found within the coding region of ORF 14 (gC). From there, the investigators compared WT gC to a gC engineered with disrupted G-quadruplexes both in transient transfection experiments and then further via recombinant viruses. Overall, the evidence suggests that G4 formation within the gC coding sequences suppresses gC expression and that disruption of these motifs can impart an observable phenotype on virus growth in culture.

**Part II – Major Issues: Key Experiments Required for Acceptance**

Reviewer #1: -It is not clear how the authors chose the viral strains and bacmids used in the study. The strains pOKA and YC01 and YC02 were not introduced and a better description would help the reader.

-The performed research on Putative G-quadruplex-forming sequences (PQS) on the genome and the choice to consider also bulged sequences and PQS with no loops is interesting: please add a summary table that categorizes the PQS.

-How many bulges per sequence were considered, were there sequences with more than one bulge?

-The authors report a partial conservation analysis on VZV genomes, taking into consideration another strain. A conservation analysis performed on all VZV genomes in GenBank needs to be performed. Add also a detailed description on PQSs location in the genome (i.e. promoter, coding, UTR etc) (partial information is contained in the S1 Appendix file).

-The authors noticed that many PQSs are located in the internal and terminal repeats. Which genomic features are contained in those regions? Promoters? Enhancers? Origins of replication?

-The final sentence of “Distribution of G4 motifs in R2” (page 7, lines 165-166) is not clear. Figure 2b shows that mutations are on non-G nucleotides but the sentence states the opposite. Can the authors clarify what was the conveyed message?

-The authors mutated the G residues (Page 7, lines 174) to T o A, but it is not explained why these nucleotides were chosen. Were all mutations aligned with the wt coding sequence (as shown in Fig S2 C), to state that the coding potential was not disrupted? Can the authors add new supplementary material on these analyses?

-Page 8, line 184, the authors define the sequence with a positive peak around 280 nm and a negative peak around 245 nm as an unusual structure. Are there differences between the recorded spectra and those provided by a random nucleotide unstructured sequence?

-Figure 4 is very difficult to interpret (too many spectra on the same graph), a better one should be provided.

-Page 8, lines 205-210, may the authors comment on the thermal stability and further G4-ligand stabilization of sequences taking into consideration the presence of bulges, short and long loops? Is there a trend in PDS binding and stabilization?

-pSG5 vector is used for different studies, it is not clear which promoter drives gC expression? Are there PQS in the expressing promoter?

-Please add a scheme showing the mutations inserted in the gC sequence

-Figure 7 A, why do the authors perform an IP prior to WB? What are the bands around 140 KDa? If they were specific HA targets, why were they not concentrated by the IP passage? There is a band corresponding to gC-HA also in the first lane of EV (empty vector): can the authors comment on that? Panels B and C, was a normalization formula used (B-actin as reference is stated in the methods section)?

-Figure 8 panel b, gC protein is indicated at a different height with respect to the band shown in Fig. 7.

-In the last result section, (G4 formation suppresses gC expression at the transcriptional level during virus infection) the infection assays are described in a very unclear way (page 10, lines 268-271), please carefully revise all this part.

-The authors state that the virus expressing the mutated gC, which is expressed earlier and more abundantly than the wild type counterpart, attenuates the cell to cell spread in plaque reduction assays. The plaque size may depend on various factors. Please analyze and show the overall replication kinetics of the mutated virus, to understand whether the gC G4-related mutation hampers or exacerbates VZV infection. Moreover, the viral titres generated by the two viruses (wild type and mutated) would give information of the viral infectivity in human cells.

-Are there circulating strains bearing the studied mutation?

Reviewer #2: They then assess the effect of mutating the g Quad elements on gene expression from a strong transfection based plasmid promoter and from a recombinant virus This raises my Major concern. In both transiently infected and recombinant infected cells, they report the expression of gC using a tagged protein. The gC protein detected in infected cells and from transiently expressed protein is far too small for fully mature gC. Multiple authors have shown the major predominant form of gC is 100-110KD, and is bigger than gE. gC contains 40KD of additional post translational modifications. However the gC in the blots in figure 8b and d, as well as in 7A, all show a protein at less than 75Kd. I do not know why, but one reason is that the C terminal HA tag is affecting gC processing; or that the mutations made are affecting translation of gC. Whatever the basis, all the subtle virus phenotypes ( and they are subtle, as gC is not required for VZV growth; could all be a result of the errant processing of the protein. As such, this must be resolved for the data to be of meaning. Blots are not quantified in anyway and need to be

2.The blots for gC expression are poor and not easily visible. In contrast blots for gE are way overexposed in figure b and d. these need to be repeated and fully quantified

3. The RNA for gE is different for the mutant and the parent/repaired virus (and apparently significant) (8E) no explanation is provided.

Reviewer #3: (No Response)

**Part III – Minor Issues: Editorial and Data Presentation Modifications**

Reviewer #1: -In the introduction, the infection linked to VZV was classified as an early childhood, but nothing is said about the worldwide distribution of VZV-positive persons and about the results of the worldwide vaccination campaign (so the actual VZV circulation in the population). The social health burden brought about Varicella and its reactivation (shingles) should be more emphasised.

-What is DW? Is it the vehicle used in controls? This is not clearly stated

Reviewer #2: 4.A few spellers including the lack of m last lane figure 6. Also “strand” versus “stand” in a couple of places Also the references lack capitalizations and occasionally some reference details

5.Methods lack how plaques were quantified, and measured. No details of the imaging process used. Studies were done in mewo cells in which plaques are invariably asymetric

Reviewer #3: Type page 5 line 112, should be “disrupted”

Fig. 7: The difference in RNA levels is about 2-fold between WT and G4m, yet protein levels seem to be substantially more than that. The investigators should provide additional explanations, addressing such issues as whether G4 might affect mRNA stability or translation efficiency to account for this. An additional consideration to address is whether WT versus G4m gC mRNA are processed and transported from the nucleus to the cytoplasm with equal efficiency.

Fig. 8: Within the context of the virus, mRNA levels for gC seem to be concordant with protein expression (in contrast to the transfection experiments). The investigators should provide some speculation to explain these apparent discrepancies.

Some VZV strains apparently have varying numbers of 42bp repeats within their gC coding regions. Additional commentary should be provided on whether the number of G4 repeats might have additional positive/negative effects on gC expression.

PLOS authors have the option to publish the peer review history of their article (what does this mean?). If published, this will include your full peer review and any attached files.

Reviewer #1: No

Reviewer #2: No

Reviewer #3: No
---

## [Decision Letter · Decision Letter 1]

29 Dec 2022

Dear Dr. Ahn,

Thank you very much for submitting your manuscript "G-quadruplexes formed by Varicella-Zoster virus reiteration sequences suppress expression of glycoprotein C and regulate viral cell-to-cell spread" for consideration at PLOS Pathogens. As with all papers reviewed by the journal, your manuscript was reviewed by members of the editorial board and by several independent reviewers. The reviewers appreciated the attention to an important topic. Based on the reviews, we are likely to accept this manuscript for publication, providing that you modify the manuscript according to the review recommendations.

The reviewers are impressed with the revised manuscript and ask only for some small changes to the text. We invite you to make the changes you see fit and resubmit the manuscript for quick, editorial review.

Sincerely,

Robert F. Kalejta

Academic Editor

PLOS Pathogens

Blossom Damania

Section Editor

PLOS Pathogens

Kasturi Haldar

Editor-in-Chief

PLOS Pathogens

orcid.org/0000-0001-5065-158X

Michael Malim

Editor-in-Chief

PLOS Pathogens

orcid.org/0000-0002-7699-2064

The reviewers are impressed with the revised manuscript and ask only for some small changes to the text. We invite you to make the changes you see fit and resubmit the manuscript for quick, editorial review.

Reviewer Comments (if any, and for reference):

Reviewer's Responses to Questions

**Part I - Summary**

Reviewer #1: Chung and co-authors submitted a revised manuscript dealing with the presence of G-quadruplexes (G4s) in the genome of the Varicella Zoster virus (VZV).

We thank the authors for considering and addressing the major issues raised in the first revision.

The information added in the introduction and the prediction and distribution of PQSs were appreciated. The initial part of the manuscript is now clearer and more complete.

Reviewer #2: the authors have done a thorough and remarkable job and have made the work far more convincing than the first version. figures have been clarified and better explained.

indeed i am ready to almost reccomend it being accepted for publication.

the article needs a spellcheck, as quite a few of the revised comments contain spellers or incorrect grammatical wordings. for a couple of examples see iines 95 global incident case

ln 169 lin 195 and other places. in s1 fig should be figure S1

line 266 the unmodified from of

265 how do they know the other forms are glycosylated?

.

Reviewer #3: (No Response)

**Part II – Major Issues: Key Experiments Required for Acceptance**

Reviewer #1: none required

Reviewer #2: no mJOR ISSUES

Reviewer #3: (No Response)

**Part III – Minor Issues: Editorial and Data Presentation Modifications**

Reviewer #1: Still, there are minor issues with the new data:

- A summary table for all the analyzed PQS, reporting CD data on topology, and stability and the results of the TDS would help the reader to have an overview of all sequences and in particular of those reported with atypical CD spectra.

- The presence of a table/figure panel reporting the aminoacidic changes caused by the G mutations would help the reader. Moreover, the authors should comment on how the changed aminoacids may influence protein folding and may contribute to the phenotypic changes of the virus. Is this new protein degraded faster than wt ORF14? which protein domain is involved in aminoacidic changes? Does mORF14 accumulate in cell granules, changing the ability of VZV to replicate? Please complete and discuss

- The authors conclude that the lack of G4s changes the VZV phenotype: this information is not sustained by the experimental data. mORF14 has aminoacidic changes that may cause viral cycle changes, this aspect needs to be commented.

Reviewer #2: spellers and ways of saying things and Figures need to be corrected

Reviewer #3: (No Response)

PLOS authors have the option to publish the peer review history of their article (what does this mean?). If published, this will include your full peer review and any attached files.

Reviewer #1: No

Reviewer #2: **Yes: **Paul R. Kinchington

Reviewer #3: No

Figure Files:

Data Requirements:

Reproducibility:

References:

---

## [Editor Report · Decision Letter 2]

2 Jan 2023

Dear Dr. Ahn,

We are pleased to inform you that your manuscript 'G-quadruplexes formed by Varicella-Zoster virus reiteration sequences suppress expression of glycoprotein C and regulate viral cell-to-cell spread' has been provisionally accepted for publication in PLOS Pathogens.

Best regards,

Robert F. Kalejta

Academic Editor

PLOS Pathogens

Blossom Damania

Section Editor

PLOS Pathogens

Kasturi Haldar

Editor-in-Chief

PLOS Pathogens

orcid.org/0000-0001-5065-158X

Michael Malim

Editor-in-Chief

PLOS Pathogens

orcid.org/0000-0002-7699-2064
---

## [Editor Report · Acceptance letter]

9 Jan 2023

Dear Dr. Ahn,

We are delighted to inform you that your manuscript, "G-quadruplexes formed by Varicella-Zoster virus reiteration sequences suppress expression of glycoprotein C and regulate viral cell-to-cell spread," has been formally accepted for publication in PLOS Pathogens.

Best regards,

Kasturi Haldar

Editor-in-Chief

PLOS Pathogens

orcid.org/0000-0001-5065-158X

Michael Malim

Editor-in-Chief

PLOS Pathogens

orcid.org/0000-0002-7699-2064